# Characterization of the Ergosterol Biosynthesis Pathway in Ceratocystidaceae

**DOI:** 10.3390/jof7030237

**Published:** 2021-03-22

**Authors:** Mohammad Sayari, Magrieta A. van der Nest, Emma T. Steenkamp, Saleh Rahimlou, Almuth Hammerbacher, Brenda D. Wingfield

**Affiliations:** 1Department of Biochemistry, Genetics and Microbiology, Forestry and Agricultural Biotechnology Institute (FABI), University of Pretoria, Pretoria 0002, South Africa; vandernestm@arc.agric.za (M.A.v.d.N.); emma.steenkamp@fabi.up.ac.za (E.T.S.); almuth.hammerbacher@fabi.up.ac.za (A.H.); brenda.wingfield@fabi.up.ac.za (B.D.W.); 2Department of Plant Science, University of Manitoba, 222 Agriculture Building, Winnipeg, MB R3T 2N2, Canada; 3Biotechnology Platform, Agricultural Research Council (ARC), Onderstepoort Campus, Pretoria 0110, South Africa; 4Department of Mycology and Microbiology, University of Tartu, 14A Ravila, 50411 Tartu, Estonia; Saleh.Rahimlou@ut.ee

**Keywords:** ergosterol, Ceratocystidaceae, terpenes, biosynthetic gene cluster

## Abstract

Terpenes represent the biggest group of natural compounds on earth. This large class of organic hydrocarbons is distributed among all cellular organisms, including fungi. The different classes of terpenes produced by fungi are mono, sesqui, di- and triterpenes, although triterpene ergosterol is the main sterol identified in cell membranes of these organisms. The availability of genomic data from members in the Ceratocystidaceae enabled the detection and characterization of the genes encoding the enzymes in the mevalonate and ergosterol biosynthetic pathways. Using a bioinformatics approach, fungal orthologs of sterol biosynthesis genes in nine different species of the Ceratocystidaceae were identified. Ergosterol and some of the intermediates in the pathway were also detected in seven species (*Ceratocystis manginecans*, *C. adiposa*, *Huntiella moniliformis*, *Thielaviopsis punctulata*, *Bretziella fagacearum*, *Endoconidiophora polonica* and *Davidsoniella virescens*), using gas chromatography-mass spectrometry analysis. The average ergosterol content differed among different genera of Ceratocystidaceae. We also identified all possible terpene related genes and possible biosynthetic clusters in the genomes used in this study. We found a highly conserved terpene biosynthesis gene cluster containing some genes encoding ergosterol biosynthesis enzymes in the analysed genomes. An additional possible terpene gene cluster was also identified in all of the Ceratocystidaceae. We also evaluated the sensitivity of the Ceratocystidaceae to a triazole fungicide that inhibits ergosterol synthesis. The results showed that different members of this family behave differently when exposed to different concentrations of triazole tebuconazole.

## 1. Introduction

Fungi produce a large variety of terpenoids that form part of a structurally and functionally diverse class of natural compounds [1]. They are involved in an array of biological processes ranging from those needed for the adaptation to particular environmental niches to those needed for the interaction with other organisms [2]. Despite this variety, all terpenoids are made from simple five-carbon precursor molecules to form compounds containing two or more isoprene units [1]. The terpenoid compounds of fungi can have two, three, four or six isoprene units, respectively referred to as monoterpenes, sesquiterpenes, diterpenes and triterpenes [3]. Most known terpene synthases of fungi catalyze the formation of sesquiterpenes, and only a few diterpene and triterpene synthases are known, and no known *bona fide* monoterpene synthases [1].

Although steroids are a major group of natural triterpenoids [4], fungi lack a large diversity of these compounds [1]. In most cases, ergosterol (24-methylcholesta-5, 7, 22-trien3b-ol) is their major sterol [5], with the main exception being primitive fungi such as those in the Chytridiomycota where cholesterol (cholest-5-en-3β-ol) is the main sterol [6]. These sterols are primarily found in cell membranes where it has diverse functions including processes essential for growth and development, the regulation of cell wall permeability, and adaptation to stress [7].

The processes involved in ergosterol and cholesterol biosynthesis have been elucidated using studies on *Saccharomyces cerevisiae*, as well as various fungi [6,7,8,9]. In general, the first set of steps involves the synthesis of farnesyl pyrophosphate (FPP; a linear terpenoid with 15 carbons) from acetyl-coenzyme A (acetyl-CoA), after which the next set of catalytic steps produce squalene (a linear terpenoid with 30 carbons) from two molecules of FPP, with the final set of steps involving the conversion of squalene to lanosterol (a tetracyclic terpenoid with 30 carbons) and its modification at various positions to produce either ergosterol or cholesterol [10]. However, numerous variations are known among fungal species [11], especially in the later stages of the biosynthesis pathway [5]. Additionally, in some species, the pathway partially causes the production of other final products and no ergosterol [5]. Due to its role in biology, distinctive structural characteristics and the unique processes of its biosynthesis, ergosterol is regarded as an ideal target for fungicides [12]. For example, pathogenic fungi may be controlled using azole compounds that inhibit lanosterol 14-α-demethylase, thereby blocking a key step in the conversion of lanosterol to ergosterol [13]. However, azole resistance in fungi has been reported, and the mechanisms allowing this include detoxification transporters, amino acid substitutions in lanosterol 14-α-demethylase protein binding sites and overexpression of the genes encoding lanosterol 14-α-demethylase [13]. Therefore, understanding ergosterol biosynthesis and potential resistance is important in the management of fungal pathogens.

Here we studied ergosterol and terpene biosynthesis in the very important plant pathogenic fungal family Ceratocystidaceae (Phylum Ascomycota, Order Microascales). Despite their economic importance, our knowledge regarding ergosterol biosynthesis or the production of other terpenoids in the Ceratocystidaceae is limited to only a few studies. Ergosterol was reported as the only sterol identified from *Bretziella fagacearum* (previously known as *Ceratocystis fagacearum*) using thin-layer chromatography and mass spectrometry [14]. In another study, the cyclic monoterpene alcohol isopulegol was isolated and identified from liquid cultures of *Endoconidiophora coerulescens* (previously known as *Ceratocystis coerulescens*) [15].

The overall goal of this study was to explore the potential of Ceratocystidaceae to produce ergosterol and other terpenoids. We accordingly identified genes involved in terpenoid biosynthesis using publicly available genome sequences for 23 species from the Ceratocystidaceae (Table 1). We then used gas chromatography-mass spectrometry (GC-MS) analysis to evaluate the production of intermediates of the ergosterol biosynthesis pathway in seven representatives of the family (i.e., *C. manginecans*, *C. adiposa*, *Huntiella moniliformis*, *Thielaviopsis punctulata*, *Br. fagacearum*, *E. polonica* and *Davidsoniella virescens*). Finally, we evaluated the sensitivity of Ceratocystidaceae to an azole fungicide. The findings of this study increase our knowledge of ergosterol biosynthesis in Ceratocystidaceae, which can be applied for developing new antifungal compounds targeting this pathway. The results of the study thus provide a valuable starting point for future studies, particularly regarding the control of pathogenic species Ceratocystidaceae.

## 2. Materials and Methods

### 2.1. Fungal Genomes and Cultures

We used the available genomic sequences from 23 species from eight genera in the Ceratocystidaceae (Table 1). These included *Ceratocystis* (*C. manginecans*, *C. fimbriata, C. eucalypticola, C. harringtonii, C. smalleyii*, *C. albifundus*, *C. platani* and *C. adiposa*), *Huntiella* (*H. moniliformis*, *H. decipiens, H. bhutanensis, H. omanensis* and *H. savannae*), *Thielaviopsis* (*T. punctulata* and *T. musarum*), *Bretziella* (*Br. fagacearum*), *Endoconidiophora* (*E. polonica* and *E. laricicola*), *Ambrosiella* (*A. xylebori*), *Davidsoniella* (*D. virescens*, *D. neocaledoniae* and *D. australis*) and *Berkeleyomyces* (*Be. basicola*). These genomes were previously shown to have high levels of completeness [16,17], suggesting that they would be useful for the identification of putative genes in involved in terpenoid biosynthesis.

For the sterol detection and quantification studies, seven representative species were used. These were *C. manginecans*, *C. adiposa*, *H. moniliformis*, *T. punctulata*, *Br. fagacearum*, *E. polonica* and *D. virescens*. The entire set of 23 fungi were also used for the in vitro fungicide sensitivity assay. The isolates were routinely cultured at 25 °C in the dark on malt extract agar (MEA) medium (2% Bacto™ malt extract [BD BioSciences, San Jose, CA USA], 2% Difco™ agar [BD BioSciences]). All isolates are available from the culture collection of the Forestry and Agricultural Biotechnology Institute (FABI) fungal.

**Table 1 jof-07-00237-t001:** Isolates numbers and genome sequence information for the species used in this study.

Species	Isolate Number ^a^	GenBank Accession Number	References
*B. fagacearum*	CMW2656	MKGJ00000000	[18]
*C. adiposa*	CMW2573	LXGU00000000	[19]
*H. moniliformis*	CMW10134	JMSH00000000	[20]
*T. punctulata*	BPI 893173	LAEV00000000	[21]
*D. virescens*	CMW17339	LJZU000000000	[22]
*D. neocaledoniae*	CMW 225392	RHDR00000000	[16]
*D. australis*	CMW 2333	RHLR00000000	[17]
*E. polonica*	CMW20930	LXKZ00000000	[19]
*A. xylebori*	CBS110.61	PCDO00000000	[23]
*Be. basicola*	CMW49352	PJAC00000000	[24]
*C. manginecans*	CMW17570	JJRZ00000000	[20]
*C. fimbriata*	CMW 15049	APWK00000000	[25]
*C. eucalypticola*	CMW 11536	LJOA00000000	[22]
*C. harringtonii*	CMW 14789	MKGM0000000	[18]
*C. smalleyii*	CMW 14800	NETT01000000	[24]
*C. albifundus*	CMW 13980	JSSU000000000	[26]
*E. laricicola*	CMW 20928	LXGT00000000	[19]
*H. decipiens*	CMW 30855	NETU00000000	[27]
*H. bhutanensis*	CMW 8217	MJMS00000000	[18]
*H. omanensis*	CMW 11056	JSUI000000000	[26]
*H. savannae*	CMW 17300	LCZG00000000	[28]
*T. musarum*	CMW 1546	LKBB00000000	[22]
*C. platani*	CF0	LBBL00000000	[29]

^a^ Isolates with CMW numbers may be obtained from the culture collection of the Tree Protection Cooperative Programme (TPCP), Forestry and Agricultural Biotechnology Institute (FABI), University of Pretoria, Pretoria, South Africa. Those with CBS and BPI numbers may be obtained from Centraalbureau voor Schimmel cultures, CBS Fungal Biodiversity Centre and the US National Fungus Collections, Systematic Botany and Mycology Laboratory, Maryland, U.S.A.

### 2.2. Identification of Terpene Biosynthesis Genes and Gene Clusters

For identifying potential terpenoid biosynthesis genes and gene clusters, the respective genome sequences were submitted, to antiSMASH [30]. The Open reading frames (ORFs) identified were further examined using the InterPro web portal [31], after which signature sequences were inferred with MOTIF (http://www.genome.jp/tools/motif/ accessed on 5 January 2021). The contigs identified by antiSMASH to contain terpene biosynthesis genes and gene clusters were also examined manually. For this purpose, any ORF in approximately 15kilobase pair upstream and downstream of the predicted core gene (i.e., a terpene synthase coding gene) were screened against the nonredundant protein database of the National Centre for Biotechnology Information (NCBI; http://www.ncbi.nlm.nih.gov accessed on 5 January 2021) using BLASTp and tBLASTn searches (Expect value [E] ≤ 10^−6^) to find their orthologs.

### 2.3. Ergosterol Biosynthesis Pathway Prediction

The GhostKoala tool of the Kyoto Encyclopedia of Genes and Genomes (KEGG) [32] was used to add metabolic annotations to all of the genes encoded by each genome. The ergosterol biosynthesis pathway in Ceratocystidaceae was then predicted using the KEGG Mapper—Search&colour Pathway tool (http://www.genome.jp/kegg/tool/map_pathway2 accessed on 5 January 2021). These analyses were done using the whole genome sequences of all of the fungi included (Table 1), with the only exception being *A. xylebori* because of an assembly problem that caused incorrect prediction of the pathway. As a reference, we used the genome of *Fusarium graminearum* (GenBank accession number AACM00000000). The annotation of homologous genes from each species was checked using both AUGUSTUS [33] and Fgenesh [34].

The genome sequences included in this study were also interrogated for the presence of putative genes involved in the production of the intermediate mevalonate and of ergosterol using BLASTn analyses (E ≤ 10^−6^) in CLC Genomic Workbench version 11.0.1 (Qiagen Bioinformatics, Aarhus, Denmark). The gene sequences used in this analysis included those that are part of the mevalonate pathway, namely genes coding for *ERG8* (EWZ41759) from *Fusarium oxysporum*, *ERG10* (EDV11184 and EDN61111) from *S. cerevisiae*, *ERG12* (EWZ48657) from *F. oxysporum*, *ERG13* (PTB67186) from *Trichoderma citrinoviride*, *ERG20* (ESU12927) from *F. graminearum*, isopentenyl-diphosphate delta-isomerase (ESU16336) from *F. graminearum* and diphosphomevalonate decarboxylase (ESU17136) from *F. graminearum.* Those that are part of the ergosterol biosynthesis phase of the pathway included genes coding for *ERG1* (AAA34592) from *S. cerevisiae*, *ERG2* (RKL43533), ERG5 (RKL44214) and *ERG7* (RBA08815) from *F. proliferatum*, *ERG3* (PCD36290), *ERG4* (EFX01240) and *ERG6* (ESU10532) from *F. graminearum*, *ERG9* (ABX64425) and *ERG25* (KLO95654) from *F. fujikuroi*, *ERG11* (DAA06695), ERG24 (EDN62547) and *ERG27* (EDN59646) from *S. cerevisiae*.

### 2.4. Intron/Exon Architecture of the Ceratocystidaceae ERG Genes

The intron and exon positions of putative ERG genes in the genomes included in this study were manually confirmed using CLC Genomic Workbench and Geneious version 11.1.5 (https://www.geneious.com/geneious/ accessed on 5 January 2021). The intron–exon boundaries for the genes in *H. moniliformis* and *C. fimbriata* were also confirmed by mapping publicly available transcripts [35,36] to the different genes of these two species. For this purpose, the available RNAseq reads were quality filtered in CLC Genomics Workbench version 9.1.1.0 using a read length over 300 bp and a Phred quality score of below 20 (Q ≤ 0.01). The resulting RNASeq data for each of the isolates were then mapped to the different genes identified, using the RNA-legacy tool in CLC Genomics Workbench with a minimum similarity fraction of 0.8 and minimum length fraction of 0.5.

### 2.5. Phylogenetic Analysis of the Putative ERG11 and ERG13 Genes of the Ceratocystidaceae

Genes encoding hydroxymethylglutaryl-CoA synthase (*ERG13*) and lanosterol 14-α-demethylase (*ERG11*) obtained from the 23 genomes used in this study, were subjected to phylogenetic analysis. These genes are highly conserved, present in all eukaryotic kingdoms, and are believed to have evolved prior to the divergence of most eukaryotic families [37]. Accordingly, the datasets analysed included the respective *ERG11* and *ERG13* sequences from all of the Ceratocystidaceae genomes analysed, together with previously sequenced genes acquired from the top blast hits in the NCBI database. For outgroup purposes, *ERG11* sequences from *Ustilago hordei* (CCF50451), *Sporisorium reilianum* (CBQ68278), *Malassezia vespertilionis* (PKI85412) and *Malassezia sympodialis* (XP_018740314) were used. For *ERG13*, the sequences from *Wallemia mellicola* (XP_006959548) and *Ustilago maydis* (XP_011392062) served as outgroups.

For both the datasets, alignments based on amino acid residues were performed using MAFFT (Multiple *Alignment using* Fast Fourier Transform; [38]) with default parameters. From these datasets, phylogenetic trees were inferred using the MEGA7 package [39] and the best-fit substitution models indicated by ProtTest [40]. These were JTT+I+G for *ERG11* and LG+I+G+F in the case of *ERG13*) Branch support was estimated using the same model parameters and 1000 pseudo replicates.

### 2.6. Sterol Detection and Quantification

Lipid extraction from fungal cultures was done as described before [41]. Briefly, this involved inoculation of 200 mL of malt extract broth (MEB; 2% Bacto™ malt extract [BD BioSciences, San Jose, CA, USA]) with a small block of agar overgrown with mycelium and incubation for 14 days at 25 °C in the darkness with shaking at 150 rpm (222DS Benchtop Shaking Incubator; Labnet international, Edison, NJ, USA). Fungal tissue was then collected by filtration through filter paper (Whatman, Maidstone, United Kingdom), washed with distilled water, and freeze-dried. Sterols were then extracted from 50 mg lyophilized mycelium by saponification at 80 °C for 90 min in a 500 µL solution containing 10% (*w/v*) KOH in methanol. To this mixture, 250 µL distilled water was added and left to cool to 25 °C. The solution was then subjected to three rounds organic extraction using 1 mL hexane. The pooled hexane fractions were evaporated to dryness under a stream of nitrogen and dissolved in 100 µL methanol.

The samples were derivatized using 50 µL N-methyl-N-trimethylsilyl-trifluoroacetamide (Macherey-Nagel, Dueren, Germany) for 2 h at 40 °C before analysis using a gas chromatograph attached to a quadrupole mass spectrometer with an electron impact ion source (Agilent 5973 6890 GC MS, Agilent, Santa Clara, USA). A standard 30 m HP-5ms capillary column (Agilent) was used with a constant flow of helium of 1 mL min^−1^. Sample (1 µL) was injected with a 1:10 split ratio. The injector temperature was kept at 230 °C. The oven temperature was ramped from an initial 70 °C to 320 °C at a constant rate of 6 °C min^−1^, with a final hold time of 3 min. The mass spectrometer was operated in full scan mode with a mass range of 50–350 m z^−1^. The ion source was kept at 70 eV at 250 °C.

Compounds were identified by spectral comparisons with the NIST library version 12 (National Institute for Standards and Technology, Boulder, USA), retention indices and when available, retention times of pure standards. Peaks were integrated and quantified relative to a calibration curve produced for cholesterol. All experiments were done in three independent replications. Data were imported into the open-source R-based statistics program Metaboanalyst (https://www.metaboanalyst.ca/faces/home.xhtml accessed on 5 January 2021), normalized using the natural logarithm. Hierarchical clustering was performed with the hclust function in the R package using Euclidean distance as the similarity measure and the Ward’s linkage clustering algorithm.

### 2.7. In Vitro Azole Sensitivity

The sensitivity of the fungi to tebuconazole (1-(4-Chlorophenyl)-4,4-dimethyl-3-(1*H*-1,2,4-triazol-1-ylmethyl)-3-pentanol, 250 g/L, Bayer, Leverkusen, Germany) was examined using the mycelial inhibition technique. Sensitivity to the compound was evaluated by transferring a 4-mm mycelial block was obtained from the growing edge of a 1-week-old MEA culture to the center of a Petri plate containing MEA medium supplemented the fungicide. For this purpose, tebuconazole concentrations of 75, 50, 25, 12.5, 6.25, 3 and 1.5 µg L^−1^ were used. The inoculated plates were then incubated for 7 days at 25 °C in the dark after which fungal colony diameters were measured. Four replicates were considered per concentration.

All measurements were subjected to one-way analysis of variance (ANOVA). An individual analysis was performed for each dependent variable and each fungal species on the repeated dataset. Assumption of normality and homogeneity of variance were assessed prior to the analysis. Means were compared using Tukey’s test at a statistical significance of *p* ≤ 0.05. The analysis was conducted by the statistics program R using “agricolae” package [42].

## 3. Results

### 3.1. Identification of Terpene Biosynthesis Genes and Gene Clusters

AntiSMASH detected two different terpenoid biosynthesis gene clusters in all the genomes examined in this study (Figure 1). The first cluster was part of the ergosterol biosynthesis pathway, because it contained genes encoding a steroid-binding protein, DnaJ, Squalene synthase, delta 24-sterol reductase, a protease and a mannoyl transferase coding genes (Figure 1A; Appendix A). The second cluster included a core gene encoding geranylgeranyl pyrophosphate synthetase [43], which was flanked by genes encoding a cell division protein, MFS transporter, peroxisomal membrane protein, arsenite transmembrane transporter and a hypothetical protein (Figure 1B; Appendix A). In *H. moniliformis* this second cluster was on the same contig, but 50 kb away, from a gene encoding lanosterol synthase (a key gene in ergosterol biosynthesis pathway [44]). In all other members of the Ceratocystidaceae, the lanosterol synthase encoding gene was located on a separate contig.

Even though antiSMASH analysis did not identify other genes in the ergosterol biosynthesis pathway, investigation of the genomes using KEGG and BLASTp analysis identified additional genes involved in the pathway. These included genes coding for phosphomevalonate kinase (*ERG8*), C-14 demethylase (*ERG11*) and C8 sterol isomerase (*ERG2*) from *H. moniliformis*, *H. omanensis*, *H. bhutanensis* and *H. savannae* (Appendix A). In *H. decipiens*, genes coding for *ERG2* and *ERG8* were found on the same contig (NETU00000079) and gene coding for *ERG11* on another contig (NETU00000037). However, in all other members of Ceratocystidaceae different *ERG* genes were scattered throughout the genome and found in different contigs.

Investigation of the genomes using BLAST analysis followed by a manual curation confirmed the presence of two putative terpene synthase coding gene sequences. However, BLAST searches across all Ceratocystidaceae fungal genomes produced five positive hits for putative terpene synthase coding genes in some genomes that were not used for further analysis, due to being either too big (*E. polonica*, *E. laricicola*) or too small (*H*. *moniliformis*) to resemble a typical terpene synthase. pBLAST analysis of these genes also confirmed that they likely did not represent terpene synthase coding genes.

### 3.2. Ergosterol Biosynthesis Pathway Prediction

We were able to find the gene repertoire for ergosterol biosynthesis in different members of Ceratocystidaceae using KEGG and BLASTp analysis. Based on these analyses, all the genes that form the ergosterol biosynthesis pathway were present in the genomes included in this study (Figure 2; Appendix A). These included genes encoding the recognized enzymes involved in ergosterol biosynthesis in *S. cerevisiae* [45], beginning with Acetyl Co-A and its ultimate conversion to ergosterol [8]. In the following section, we present the gene homologues in different Ceratocystidaceae following their order in the predicted ergosterol biosynthesis pathway.

The gene homologues from the first phase of the pathway (synthesis of FPP from Acetyl Co-A) that were identified in the Ceratocystidaceae included those encoding for acetyl-CoA C-acetyltransferase, ERG13 (hydroxymethylglutaryl-CoA synthase), hydroxymethylglutaryl-CoA reductase (HMGCR), ERG12 (mevalonate kinase), ERG8 (phosphomevalonate kinase) and ERG20 (farnesyl diphosphate synthase) (Appendix A).

With regards to the second phase of ergosterol biosynthesis (synthesis of ergosterol from FPP), we found the gene coding for the conserved squalene synthase (ERG9) among all the Ceratocystidaceae examined. BLASTp analysis of squalene synthase from different Ceratocystidaceae showed a very high similarity to those of different *Ascomycetes* such as *Neurospora tetrasperma* (an average of 88–92%, XP_009853279) and *Sordaria macrospora* (an average of 87–89% XP_003345124). Overall, the Ceratocystidaceae squalene synthase protein sequences shared 87–64% amino acid similarity with those of other Ascomycota. A similar trend of high similarity (88–61%) among the Ceratocystidaceae was also observed for the next enzyme in the pathway, squalene monooxygenase (ERG1). This enzyme is responsible for the formation of oxidosqualene by epoxidation of squalene [46]. Its protein sequences in the Ceratocystidaceae were most similar to those of *Verticillium longiospermum* (an average of 64%, CRK44748), *Verticillium dahliae* (an average of 65%, PNH40870) and *Hypoxylon* sp. (an average of 66%, OTA91656).

Proteins encoded by ERG7, ERG11 and ERG24 respectively responsible for formation of lanosterol from *squalene epoxide* and the subsequent demethylation and reduction of lanosterol, to ultimately form 14-Dimethyl-lanosterol were also found in all the genomes examined. The Ceratocystidaceae *ERG7* protein sequences shared high similarity to its homologs in Colletotrichum graminicola (an average of 74%, XP_008098265), *Lemontospora prolificans* (an average of 72%, PKS07236) and *Scedosporium apiospermum* (an average of 70%, XP_016640229). The ERG11 protein sequences from the Ceratocystidaceae also shared high similarity to homologs in other fungi (an average of 76% to Lemontospora prolificans, AWO72586, 71% to *Scedosporium apiospermum*, AWO72588, 73% to *Metarhizium robertsii*, XP_007820238), while those of ERG24 shared high similarity to homologs from fungi such as *Scedosporium apiospermum* (an average of 66%, XP_016642425) and *Lemontospora* prolificans (an average of 69%, PKS10286).

Genes coding for *ERG25* (*encoding* C-4 sterol methyl oxidase), *ERG26* (*encoding* sterol 4-α-carboxylate 3-dehydrogenase) and *ERG27* (*encoding* 3-keto sterol reductase) work together as enzymatic complexes [47] and were present in all the Ceratocystidaceae genomes examined. *ERG25* sequences of *Lemontospora prolificans* (an average of 78%, PKS08950) and *Scedosporium apiospermum* (an average of 76%, XP_016640914) were most similar to those of the Ceratocystidaceae. The *ERG26* protein product is a member of the 3b-hydroxysteroid dehydrogenase family and removes the 3a-hydrogen from 4-Methyl zymosterol carboxylate, which results in the 3keto-4methyl zymosterol intermediate decarboxylation [48]. The identified *ERG26* gene shared high similarity to those encoded by other Ascomycetes, e.g., an average of 75% to *Lemontospora prolificans* (AWO72586), an average of 77% to *Scedosporium apiospermum* (AWO72588) and an average of 74% to *Metarhizium robertsii* (XP_007820238). The *ERG27* gene product catalyses formation of zymosterol from 3,keto-4,methyl zymosterol [49]. The Ceratocystidaceae *ERG27* protein sequences were most similar to those of *Colletotrichum simmondsiis* (an average of 79%, KXH40884), *Colletotrichum nymphaeae* (an average of 76%, KXH38700) and *Colletotrichum salicis* (an average of 77%, KXH38261).

*ERG6*, catalyses the C-24 methylation of zymosterol to form fecosterol [50]. The putative protein sequence of *ERG6* was most similar to species in the genus *Colletotrichum* (e.g., an average of 86% to *Colletotrichum gloeosporioides* (ELA25057), *Colletotrichum chlorophyti* (OLN94084) and *Colletotrichum incanum* (KZL83690). All the Ceratocystidaceae genomes encoded a *ERG28* gene, which also showed high similarity to the homolog in *Colletotrichum* (e.g., *Colletotrichum chlorophyti* (an average of 98%, OLN97423), *Colletotrichum orchidophilum* (an average of 95%, XP_022474034).

Homologs of the genes coding for *ERG2*, *ERG3*, *ERG4* and *ERG5* were also detected in all the Ceratocystidaceae genomes examined. The products of these genes form the final part of the ergosterol biosynthetic pathway. The gene coding for *ERG2* encodes sterol 8-isomerase that moves the double bond of the B ring of fecosterol from position 8 to position 7 to form Episterol [51]. The Ceratocystidaceae *ERG2* protein sequences were most similar to those of *Trichoderma asperellum* (an average of 68%, XP_024761716), *Trichoderma atroviride* (an average of 68%, XP_013948848) and *Chaetomium globosum* (an average of 66%, XP_001224797). *ERG3* encodes sterol C-5 desaturase that catalyses formation of the double bond (carbon–carbon bond) in episterol, resulting in the formation of the toxic 2,7,24,28-Ergosta trienol. The Ceratocystidaceae *ERG3* were similar to those of other fungi, including *Trichoderma reesei* (an average of 67%, XP_006965542), *Trichoderma citrinoviride* (an average of 66%, XP_024749895) and *Metarhizium bruneum* (an average of 69%, XP_014544386). Finally, *ERG5* and *ERG4*, respectively encoding a P-450 Sterol C-22 desaturase and a C-24 sterol reductase, are responsible for the last two steps involving formation of the C-22 carbon–carbon double bond in the sterol side chain to ergosta tetraenol and ultimate conversion of this molecule to ergosterol. The Ceratocystidaceae *ERG5* was most similar to their corresponding homologs in fungi such as *Purpureocillium lilacinum* (an average of 77%, PWI70663), *Sordaria macrospora* (an average of 75%, XP_003348687) and *Colletotrichum chlorophyti* (an average of 78%, OLN95588). In the case of *ERG4*, Ceratocystidaceae share 65–74% identity with different *Verticillium* sequences (*V. alfalfa* XP_003008950, *V. dahliae* XP_009649917 and *V. longisporum* CRK47680).

### 3.3. Intron/Exon Architecture of the Ceratocystidaceae ERG Genes

In all nine of the Ceratocystidaceae genomes in which intron–exon architecture was examined, the genes encoding *ERG11*, *ERG8*, *ERG12*, mevalonate diphosphate decarboxylase and *ERG20*, displayed similar intron–exon architectures. The only exception was *ERG10*, where *A. xylebori* and *H. moniliformis* had three introns while all the other Ceratocystidaceae had two. Mapping the RNAseq data to the *ERG10* coding gene model in these fungi confirmed the intron and exon positions and distribution patterns observed for *H. moniliformis* and *C. fimbriata*.

The intron–exon architecture in the remainder of the *ERG* genes in the ergosterol pathway of different members of Ceratocystidaceae showed more diversity. For example, similar patterns were observed in Ceratocystidaceae genes coding for *ERG1*, *ERG2* and *ERG6*, but different intron–exon architectures were found within different Ceratocystidaceae for the genes coding for *ERG4*, *ERG5*, *ERG7*, *ERG9*, *ERG13*, *ERG24* and *ERG27* (Appendix A).

### 3.4. Phylogenetic Analysis the Putative ERG11 and ERG13 Genes of the Ceratocystidacea

In this research we did the phylogeny analysis on two important genes of ergosterol biosynthesis pathways include *ERG11* and *ERG13* to shed light on their variability between different members of Ceratocystidaceae family and also to check if they follow the basic phylogenetical pattern of this family or if we can find any evidence of horizontal gene transfer among tested isolates. However, based on our analysis, Ceratocystidaceae sequences were separated into a well-supported clade in both the *ERG11* and *ERG13* trees (Figure 3 and Figure 4). Moreover, in both cases, similar groupings amongst the Ceratocystidaceae sequences were seen. These groupings are congruent with the general taxonomy of Ceratocystidaceae [52].

### 3.5. Sterol Identification and Quantification

The sterol composition of the mycelium of the eight isolates representing eight genera from the family Ceratocystidaceae grown in MEA was evaluated using GC-MS (Appendix A). Ergosterol was detected in the mycelium of all isolates where the concentrations ranged from 1.01 to 3.272 mg g^−1^ dry weight (wt) mycelia. Total ergosterol was 3.272 mg g^−1^ for *C. adiposa*, 1.679 mg g^−1^ for *Br. fagacearum*, 1.194 mg g^−1^ for *T. punctulata*, 3.127 mg g^−1^ for *H. moniliformis*, 1.01 mg g^−1^ for *C. manginecans*, 3.005 mg g^−1^ for *E. polonica* and 1.492 mg g^−1^ for *D. virescens*.

The GC-MS examination of the total sterol fraction gained from different strains of Ceratocystidaceae showed that the main sterols present were lanosterol and ergosterol. However, we also detected ergosta-7,22-dien-3-ol, eburicol, 4-methyl ergosterol, fecosterol, and ergosta-5,8-dien-3-ol. The sterol profiles between the different genera included in this study differed widely (Figure 5). For instance, *E. polonica and H. monilliformis* showed similarities in their sterol profile with high levels of ergosterol and ergosta-5,8-dien-3-ol. On the other hand, the *Ceratocystis* species, *T. punctulata* and *D. virescens* shared high levels of 4-methyl ergosterol and lanosterol.

### 3.6. In Vitro Triazole Sensitivity of Ceratocystidaceae Isolates

The different members of Ceratocystidaceae showed varying patterns of sensitivity to tebuconazole. Among all tested strains, *C. adiposa*, *Thielaviopsis* species and *Davidsoniella* species were least sensitive, whereas *Br. fagacearum* and *Endoconidiophora* species showed significantly higher sensitivity to triazole (Figure 6). This fungicide inhibited mycelial growth of *Br. fagacearum*, *Davidsoniella* and *Huntiella* species significantly more than *Ceratocystis*, *Thielaviopsis* and *Endoconidiophora* species at concentrations of 3 and 6 µg L^−1^ (Figure 6). At higher concentrations of 12.5 and 25 µg L^−1^ of Tebuconazole, inhibition was not significantly different among different strains except for *Br. fagacearum* which stopped growing at 25 µg L^−1^. However, at these concentrations, *C. adiposa* and *Thielaviopsis* species were still more resistant compared to the other species. *Endoconidiophora* species growth was completely inhibited at 50 µg L^−1^. The growth of *A. xylebori*, *Huntiella* species as well as *Ceratocystis* species (except for *C. manginecans*) has stopped at 75 µg L^−1^. The other tested Ceratocystidaceae could still grow even at 75 µg L^−1^. The list of all isolates used in the sensitivity study is shown in Appendix A.

## 4. Discussion

Although ergosterol and steroid biosynthesis pathways have been extensively examined in model fungi, little is known about these pathways in non-model fungi, including those in the economically important Ceratocystidaceae. However, the availability of genome sequences, allowed us to fill this knowledge gap. The current study presents the first report of sterols, and ergosterol, in particular, produced by Ceratocystidaceae, as well as the putative biosynthetic pathway underlying ergosterol biosynthesis in these fungi. Basic knowledge of the Ceratocystidaceae ergosterol pathway is important as it could be used to develop inhibitors with a larger range or better activity. Results of this study can aid in the management of this important group of fungi, for example, the products identified in this study can have importance in the biocontrol process.

Our results suggested that the Ceratocystidaceae representatives examined have a complete set of the enzymes known to be involved in the mevalonate and ergosterol biosynthesis pathways [8]. Since these putative genes are present in the Ceratocystidaceae genomes, as well as the genomes of other Ascomycota and in the Basidiomycota, these genes are postulated to have originated in an early common of these phyla [53]. Our finding is thus in-line with the hypothesis that genes involved in the ergosterol biosynthesis pathway are crucial for the survival of most fungal lineages of the Dikarya [7].

The Ceratocystidaceae encodes all of the gene products needed to convert Acetyl-CoA into ergosterol. Ergosterol biosynthesis is facilitated by 25 coding genes [8], where the first phase is characterized by the mevalonate pathway forming FPP, an important precursor for the biosynthesis of sterols in general, as well as dolichol, heme and prenylated proteins [10]. During this first phase, hydroxymethylglutaryl-CoA reductase (HMGCR) may catalyze the rate-limiting step, as is shown in other fungi [54]. During the second phase of ergosterol biosynthesis, ERG11 (lanosterol 14-α-demethylase) and ERG1 (squalene epoxidase) likely catalyze rate-limiting steps [54], while ERG11 might also exert a regulatory effect on the ergosterol biosynthesis in general [55]. It would be interesting to see whether all of the genes identified in Ceratocystidaceae indeed fulfill their predicted roles, and to explore the intricacies that may or may not characterize ergosterol biosynthesis in these fungi.

Comparative analysis of this pathway in 23 members of Ceratocystidaceae showed that there are no duplicated genes in ergosterol biosynthetic pathway. This is congruent with what is observed in many yeasts, such as *Candida albicans* and *S. cerevisiae* [7]. However, this is different from many other *Sordariomycetes* in some species harbour more than one copy of certain ergosterol pathway genes. For instance, two genes coding for *ERG3* were found in the genomes of *F. circinatum* and *F. graminearum* while, *Magnaporthe grisea* and *F. graminearum*, had three and two genes coding for *ERG11*, respectively [8,56]. Copy number variation is a significant source for genetic polymorphism and can lead to phenotypic diversity in the population [57]. Knowing the copy number can also help us with the functional characterization studies such as a gene knockout as genes with only one copy are the best targets for such experiments.

Like other fungi, terpenoid synthesis in the Ceratocystidaceae is carried out by genes that are clustered and co-regulated similar to that observed in other fungal biosynthetic pathways [58]. Such clustering is thought to allow for easier regulation through epigenetics and control by transcription factors, which are, in turn, crucial for the adaptation and survival of fungi under changing environmental conditions. Of the two terpenoid biosynthesis-related clusters found in all the fungal genomes included in this study, both contained the expected genes (e.g., terpene synthases or cyclase genes as core gene, along with tailoring enzymes such as cytochrome P450 monoxygenases, oxidoreductases, transferases and NAD(P)+) [3]. Where one of the clusters is likely involved ergosterol biosynthesis [46], the biological role of the other is unclear. It may be involved in the production of carotenoids or antimicrobial diterpenes, because the core gene of the first terpene cluster encodes a geranylgeranyl pyrophosphate synthetase (GGPS) that is documented to form carotenoids or antimicrobial diterpenes [59]. However, it is possible that this gene may be involved in the synthesis of geranylgeranyl for protein prenylation rather than for the synthesis of a secondary metabolite. This is because many fungi have two copies of the geranylgeranyl diphosphate synthase gene [60,61]. One of the geranylgeranyl diphosphate synthase genes in fungi is involved in protein prenylation, while the other gene product is involved in SM biosynthesis [62]. It is interesting that in *Huntiella* species, the GGPP-containing cluster was located in the vicinity of gene encoding lanosterol synthase. It might be possible that these gene clusters have an ancient origin which has been kept specifically in the genome of *Huntiella*. It is equally possible that the clustering in *Huntiella* is a more recent event.

The positional variation of introns in the *ERG gene* among different members of Ceratocystidaceae may suggest evolutionary events such as recombination or exon shuffling [63]. Positional dissimilarity (intron shuffling) was observed in the case of genes coding for *ERG4*, *ERG5*, *ERG7*, *ERG9*, *ERG10*, *ERG13*, *ERG24* and *ERG27*. In contrast, the intron positions in *ERG1*, *ERG2*, *ERG6*, *ERG8*, *ERG11* and *ERG12* were conserved in all species. Intron shuffling in *ERG11* among other fungi include *Aspergillus niger*, *Nectria haematococca*, *Fusarium sporotrichioides*, *Penicillium italicum* and *Fusarium oxosporium* [63]. The average intron size in the genes coding for *ERG11* in Ceratocystidaceae was 69 bp, which is similar to the size predicted for other genes analysed from other fungi using genome sequencing [64]. The branching pattern for both the *ERG11* and *ERG13* phylogenies produced the same overall branching patterns and the intron conservation is congruent with these phylogenies. This might be likely to put some limitations on the history of sterol biosynthesis due to these explanations.

From this study, it is clear that Ceratocystidaceae members produce ergosterol as their main sterol. Our biochemical analysis revealed that *C. manginecans* had the lowest concentration of ergosterol and *C. adiposa* had the highest compared to the other members of the family. Ergosterol content of these fungi was comparable to levels found in other fungi such as *Aspergillus amstelodami*, *Alternaria alternata* and *Aspergillus flavus* (1 to 5 mg mg^−1^ dry weight) [65]. Although, the amount of ergosterol in Ceratocystidaceae was much higher than those measured from several other fungi associated with *Xyleborus* ambrosia beetles (0.12%–0.24 µg/mg) [66], it was still much lower than that quantified for fungi such as *Aspergillus fumigatus*, *Candida albicans* and *Aspergillus flavus* (up to 14 µg/mg) [67].

In our analysis, we were unable to detect lanosterol in isolates of *Br. fagacearum* and *H. moniliformis*. We also could not identify any fecosterol in *D. virescens*. Diversity in sterol content and composition among Ceratocystidaceae is probably a result of differences amongst isolates and species, as well as differences in growth stage tested, growth temperature, and sterol isolation method [68]. Media composition was also reported to have an effect on sterol detection in fungi, but since all Ceratocystidaceae mycelia were grown in the same media, it is unlikely to be a variable for this study.

In Ceratocystidaceae, we could not detect any of the standard ergosterol pathway intermediates through zymosterol [6]. Rather, we found all intermediate sterols of fecosterol formation through eburicol. Fecosterol and eburicol were detected in some of the strains. These findings suggest that the production of fecosterol through eburicol in members of Ceratocystidaceae is the same as the pathway described for ergosterol biosynthesis in *Aspergillus* species [46].

Different members of Ceratocystidaceae showed different patterns of sensitivity to the triazole fungicide tebuconazole, indicating significant differences in ergosterol content or level of expression of the genes in the ergosterol pathway among these strains. This could be also a result of differences in the number of ABC transporters which will result in different accumulation of the fungicide in fungal cells. Nevertheless, the results presented here are congruent with the outcomes of other reports in which the survival and growth and of *C. fimbriata* on Potato Dextrose Agar medium were completely halted by the triazole propiconazole [69]. In another study by Scruggs et al. (2017 [70]) on the in vitro effect of different chemicals on *C. fimbriata*, the ergosterol inhibitor fungicide, difenoconazole, was reported to be the most effective fungicide tested. The differences in azole sensitivity among different members of Ceratocystidaceae, as seen in the present study, demonstrates the necessity for further transcriptional and population–genetics studies in these microorganisms.

## Figures and Tables

**Figure 1 jof-07-00237-f001:**
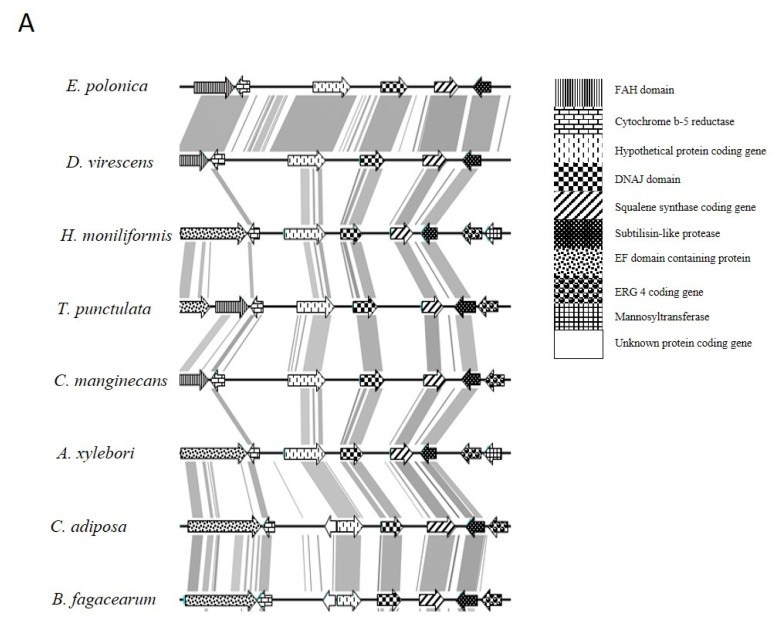
Comparison of gene content and organization of the two terpenoid biosynthesis clusters identified in different members of Ceratocystidaceae. One of these clusters (**A**) contained genes needed for ergosterol biosynthesis. The other contained a gene encoding geranylgeranyl pyrophosphate synthetase (**B**) which may be involved in protein prenylation or potentially in the production of carotenoids or antimicrobial diterpenes (Keller et al., 2005). The direction and relative size of genes are indicated by arrows. The function of putative synthetic genes is shown by different patterns. Each color/pattern represents a specific gene.

**Figure 2 jof-07-00237-f002:**
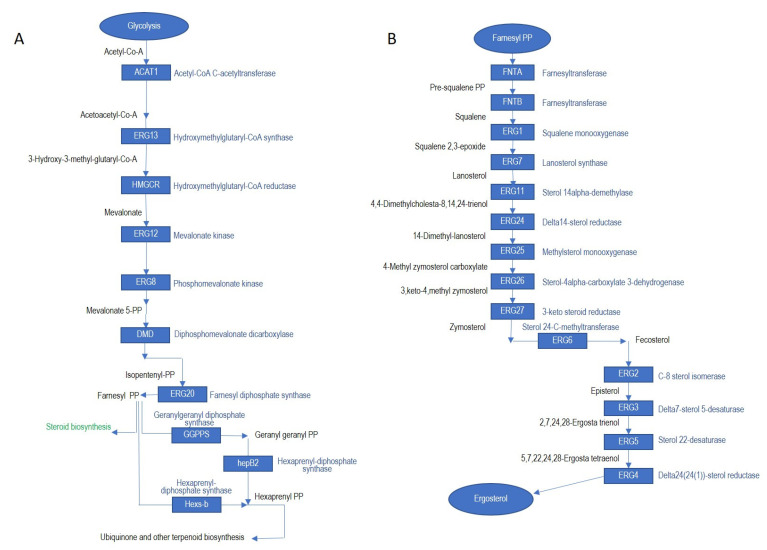
Possible ergosterol biosynthesis pathway in Ceratocystidaceae, as adapated from Bhattacharya et al. (2018). (**A**): the first part of the pathway from Acetyl-CoA to farnesyl pyrophosphate (FPP). (**B**): the last part of the pathway from FPP to ergosterol. ACAT1: Acetyl-CoA C-acetyltransferase, ERG13: Hydroxymethylglutaryl-CoA synthase, HMGCR: Hydroxymethylglutaryl-CoA reductase, ERG12: ERG12, ERG8: Phosphomevalonate kinase, DMD: Diphosphomevalonate dicarboxylase, ERG20: Farnesyl diphosphate synthase, GGPPS: Geranyl geranyl PP, hepB2: Hexaprenyl-diphosphate synthase, Hexs-b: Hexaprenyl PP, FNTA: Farnesyltransferase, FNTB: Farnesyltransferase, ERG1: Squalene monooxygenase, ERG7: Lanosterol synthase, ERG11: Sterol 14alpha-demethylase, ERG24: Delta14-sterol reductase, ERG25: Methylsterol monooxygenase, ERG26: Sterol-4alpha-carboxylate 3-dehydrogenase, ERG27: 3-keto steroid reductase, ERG6: Sterol 24-C-methyltransferase, ERG2: C-8 sterol isomerase, ERG3: Delta7-sterol 5-desaturase, ERG5: Sterol 22-desaturase, ERG4: Delta24(24(1))-sterol reductase.

**Figure 3 jof-07-00237-f003:**
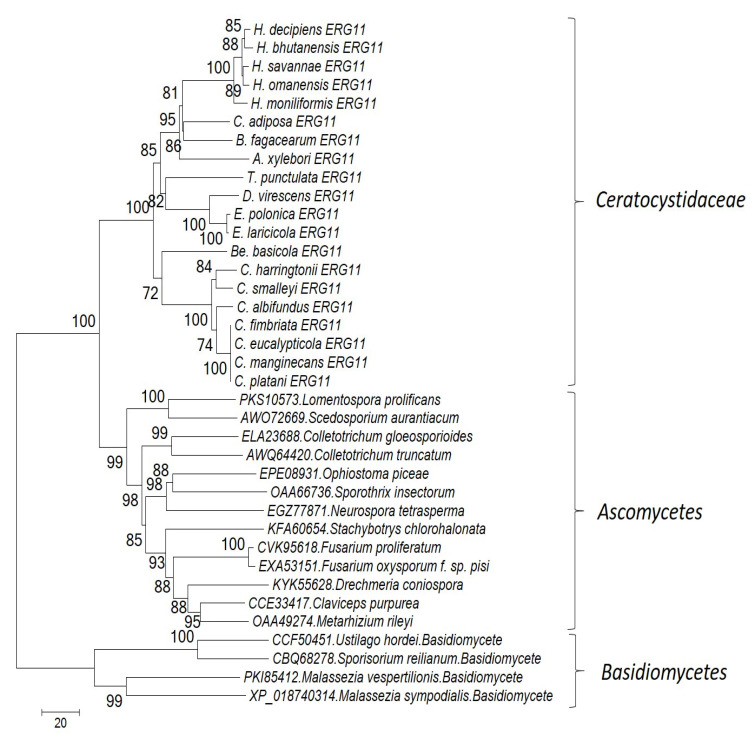
A maximum-likelihood tree of *ERG11* protein sequences from different members of Ceratocystidaceae (one or more species of the following genera: *Ceratocystis*, *Huntiella*, *Thielaviopsis*, *Bretziella*, *Endoconidiophora*, *Ambrosiella*, *Dadidsoniella* and *Berkeleyomyces*), was achieved using the MEGA7 software and substitution model LG+I+G+F. The *ERG11* from Ceratocystidaceae is shown with an underline. The evolutionary distances are in the units of the number of amino acid differences per sequence. The analysis involved 31 amino acid sequences. All positions with less than 95% site coverage were eliminated.

**Figure 4 jof-07-00237-f004:**
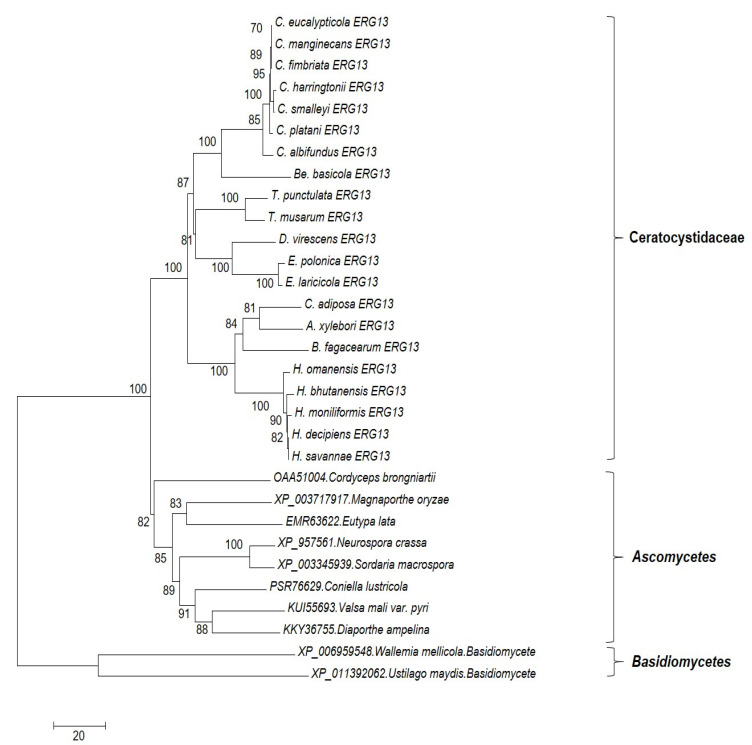
Phylogenetic tree performed on *ERG13* protein sequences. This tree was constructed inferred using the maximum likelihood method using MEGA7 software. The optimal tree is shown. The percentage of replicate trees in which the associated taxa clustered together in the bootstrap test (1000 replicates) is shown next to the branches. The analysis involved 46 amino acid sequences. All positions with less than 95% site coverage were eliminated. That is, fewer than 5% alignment gaps, missing data, and ambiguous bases were allowed at any position. There was a total of 442 positions in the final dataset. Tree includes *ERG13* proteins from one or more species of the following genera in the family Ceratocystidaceae: *Ceratocystis*, *Huntiella*, *Thielaviopsis*, *Bretziella*, *Endoconidiophora*, *Ambrosiella*, *Dadidsoniella* and *Berkeleyomyces*.

**Figure 5 jof-07-00237-f005:**
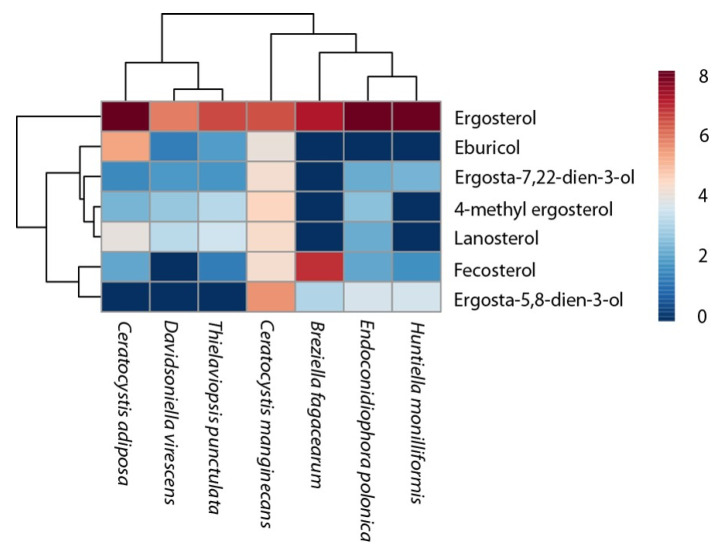
Heatmap of the sterol composition and ergosterol content in mycelia of isolates of the Ceratocystidaceae species: *C. manginecans*, *H. moniliformis*, *T. punctulata*, *B. fagacearum*, *E. polonica* and *D. virescens* grown in the MEB media for seven days. All three replicates are shown. Red and blue colours indicate higher and lower concentrations of the compounds, respectively.

**Figure 6 jof-07-00237-f006:**
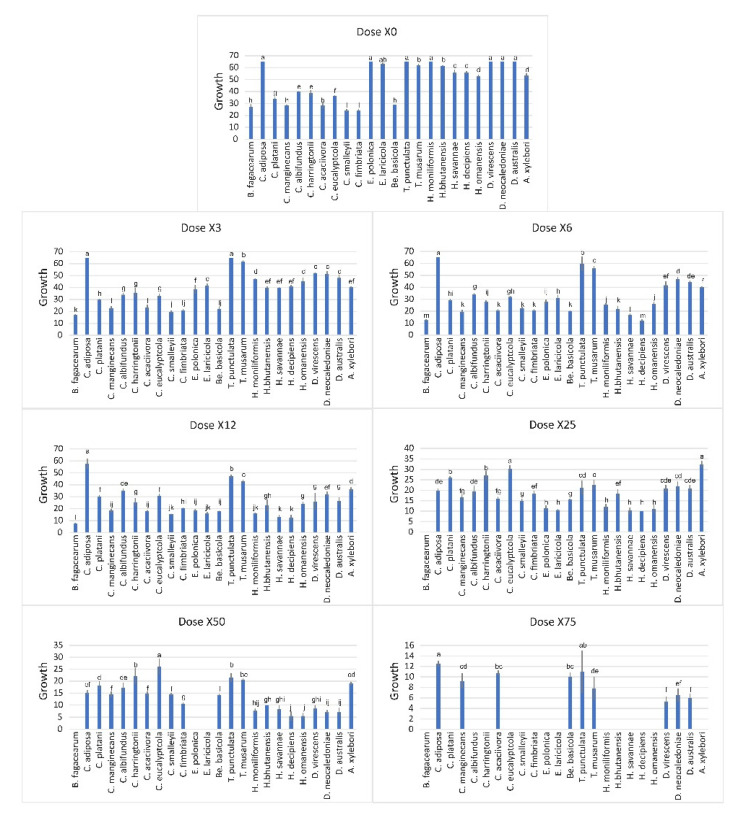
Effect of different triazole concentrations on mycelial growth of different Ceratocystidaceae including Ceratocystis (*C. manginecans*, *C. fimbriata*, *C. eucalypticola*, *C. harringtonii*, *C. smalleyii*, *C. albifundus*, *C. platani* and *C. adiposa*), Huntiella (*H. moniliformis*, *H. decipiens*, *H. bhutanensis*, *H. omanensis* and *H. savannae*), Thielaviopsis (*T. punctulata* and *T. musarum*), Bretziella (*Br. fagacearum*), Endoconidiophora (*E. polonica* and *E. laricicola*), Ambrosiella (*A. xylebori*), Davidsoniella (*D. virescens*, *D. neocaledoniae* and *D. australis*) and Berkeleyomyces (*Be. basicola*). Agar discs containing each isolate were grown for 7 days at 25 °C on MEA media amended with triazole at six different concentrations. Bars represent standard error. There were significant differences between means (*p* < 0.05) based on Tukey’s test.

## Data Availability

All data sets generated for this study are included in the manuscript and the Appendix A.

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
