# Peer review of "Characterization of the Ergosterol Biosynthesis Pathway in Ceratocystidaceae"

_jof, 2021, doi:10.3390/jof7030237_

Round 1
Reviewer 1 Report
The subject of the manuscript is very interesting and it is high need of the such studies. The experimental design is good and the results are well presented.
Author Response
Response to Reviewers’ Comments
Dear Reviewer,
Thanks for your time to review our paper. All other changes are highlighted in yellow in the text.
Reviewer 1:
This reviewer was happy with the manuscript:
Thanks for the positive feedback. We appreciate it.
Regards,
Mohammad Sayari
Reviewer 2 Report
Dear Authors,
The manuscript describe the genome mining for well-characterized ergosterol biogenesis pathway from 23 fungal strains that previously reported by Author's group. GCMS profiling was carried out to detect the presence of ergosterol and its intermediates. Finally, antifungal assay was performed using tebuconazole, an established fungicide agent.
1) The manuscript was too long, especially the introduction. The cited references have over hundreds of papers, I think that is too much considering this manuscript was submitted as article.
2) The images from figures 1, 2, 3, 4 and 6 were too small and blurred (low quality).
This work kind of predictable, ergosterol is found in cell membranes of fungi, therefore the presence of ergosterol biosynthetic gene cluster within the genomic content of investigated strain is reasonable. This is same apply to presence of ergosterol and its intermediates in these strains. The fungicide agent, tebuconazole has been marketed and used by farmers, therefore it is also reasonable that this compound has antifungal activities against the investigated strains.
Round 2
Reviewer 2 Report
Dear Authors,
The comments have addressed positively.
Author Response
Please see the attachment.
Regards,
Mohammad